# The State of the Art and Prospects for Osteoimmunomodulatory Biomaterials

**DOI:** 10.3390/ma14061357

**Published:** 2021-03-11

**Authors:** Andreea-Mariana Negrescu, Anisoara Cimpean

**Affiliations:** Department of Biochemistry and Molecular Biology, Faculty of Biology, University of Bucharest, 91-95 Splaiul Independentei, 050095 Bucharest, Romania; andreea.mariana.negrescu@drd.unibuc.ro

**Keywords:** biomaterials, bone regeneration, osteoimmunomodulation, immune response, macrophage polarization

## Abstract

The critical role of the immune system in host defense against foreign bodies and pathogens has been long recognized. With the introduction of a new field of research called osteoimmunology, the crosstalk between the immune and bone-forming cells has been studied more thoroughly, leading to the conclusion that the two systems are intimately connected through various cytokines, signaling molecules, transcription factors and receptors. The host immune reaction triggered by biomaterial implantation determines the in vivo fate of the implant, either in new bone formation or in fibrous tissue encapsulation. The traditional biomaterial design consisted in fabricating inert biomaterials capable of stimulating osteogenesis; however, inconsistencies between the in vitro and in vivo results were reported. This led to a shift in the development of biomaterials towards implants with osteoimmunomodulatory properties. By endowing the orthopedic biomaterials with favorable osteoimmunomodulatory properties, a desired immune response can be triggered in order to obtain a proper bone regeneration process. In this context, various approaches, such as the modification of chemical/structural characteristics or the incorporation of bioactive molecules, have been employed in order to modulate the crosstalk with the immune cells. The current review provides an overview of recent developments in such applied strategies.

## 1. Introduction

Annually, millions of people suffer from common bone defects caused by trauma, infection, tumor resection and pathological processes [1,2]. Even though healthy bone tissue has an extraordinary capacity for self-repair, around 10% of patients develop severe complications, such as delayed healing and non-union, leading to more expensive and often invasive treatment strategies [3]. A common method of treatment involves the use of bone grafts for the rapid restoration of larger defects/injuries, but a series of disadvantages, such as an insubstantial amount of satisfactory graft material available for use and morbidity at the donor site (e.g., nerve injury, pain, hemorrhage, infection), limits its use as a therapeutic tool for bone regeneration [4,5,6,7]. Because of the limitations associated with the conventional treatment, in the last few decades, various implantable biomaterials have been developed and tested as promising bone substitute alternatives. Their wide usability relies on their ability to act as biocompatible supports and delivery platforms for biologically active molecules, which can be easily tailored for a specific purpose (e.g., modification of the chemical and physical properties) [8,9,10,11]. However, regardless of their inert and non-toxic character, adverse immune reactions such as excessive inflammation, impairment of healing, fibrotic encapsulation and implant rejection can occur due to their exogenous nature [11,12]. The traditional method of designing implantable biomaterials focuses on developing bone substitutes that can elicit a favorable osteogenic and osseointegration process, by tailoring their mechanical and physicochemical characteristics [2,13,14]. Nevertheless, this concept does not always lead to satisfying results, with studies reporting certain inconsistencies between the in vitro and in vivo results, thus leading to the hypothesis according to which the mechanism underlying the capacity of the materials to mediate bone regeneration is known only partially and it is far more complex than it was thought [13]. For a long time, it was believed that the bone dynamics involve only cells from the skeletal system, such as osteoblasts and osteoclasts, but a more sophisticated understanding of the bone biology suggests that the osteogenic process is a result of the interplay between the skeletal and immune systems. Essential events in the bone remodeling process, such as hematopoiesis, structural support and mineralization, require intimate cooperation between the skeletal and immune systems, through numerous common regulatory molecules, such as cytokines, transcription factors, receptors and other signaling molecules [15,16,17,18]. For example, cytokines such as the transforming growth factor (TGF-β) and interleukin (IL)-4 have been reported to induce osteoblast migration, proliferation and secretion of the extracellular matrix (ECM) in the early stage of cell differentiation [19,20], while tumor necrosis factor-alpha (TNF-α) and IL-1β have the opposite effect, being involved in the inhibition of the differentiation process [21]. Moreover, the immune system has been recognized as being involved in the first stage of the natural healing process and studies have shown that treatment with anti-inflammatory drugs leads to impaired fracture healing [22,23,24,25]. This crosstalk between the immune and skeletal systems was first identified in a series of pioneering studies on osteoclast-activated factors derived from immune cells in the 1970s [26]. However, almost 30 years later, the term “osteoimmunology” was first coined and used to describe the introduction of a new direction in research and of a new interdisciplinary field [26,27]. With this advancement in bone biology, it was detrimental that a shift in the design paradigm should take place, from biomaterials capable of direct activation of cells responsible for the osteogenic process, towards biomaterials capable of modulating the local immune environment in favor of bone healing and regeneration [13,28,29]. Biomaterials endowed with immunomodulatory properties are classified as osteoimmunomodulatory biomaterials. Materials presenting favorable osteoimmunomodulatory properties are capable of inducing a suitable inflammatory response that results in the formation of new bone tissue by issuing factors from inflammatory cells capable of increasing osteogenic cell recruitment and differentiation. On the other hand, biomaterials with poor osteoimmunomodulatory properties will lead to an excessive chronic inflammatory process coupled with increased osteoclast formation, resulting in bone destruction, fibrous capsule formation and, in the end, implant failure [13,29]. Therefore, osteoimmunomodulation brings forth a promising strategy for designing bone biomaterials with multifactorial effects, such as tuning the immune system, promoting osteogenesis and regulating osteoclastogenesis [30,31,32].

In this review, we highlight the interplay between the skeletal and immune system and discuss how the specific characteristics of the biomaterials could be tailored in order to tune the immune response elicited by the implantable biomaterials for favorable tissue regeneration.

## 2. Overview of the Immune System

The immune system represents a powerful and diverse defensive tool, involved in host protection against foreign threats and body homeostasis control. The main function of the immune system is to resolve infections, repair the injured tissue and return the organism to a state of homeostasis. Its efficacy relies on the ability to exhibit a specific yet limiting, rapid and destructive response, appropriate for the inflammatory trigger [33]. At the most basic level, the human immune system can be classified into two interconnected branches, the innate immune system and the adaptive immune system, both involved in defending the organism against numerous threats, such as injuries, microbes, bacteria and toxins or any other causes [34,35]. The innate immune system is intimately connected to the inflammatory pathways and wound healing system, being capable of eliciting a non-specific immune response on contact with a foreign material or injured tissue, without previous programming [11,36]. Due to its non-specific nature, the innate immune system represents the first line of defense, being activated when certain molecules known as non-infectious damage-associated molecular patterns (DAMPs) and infectious pathogen-associated molecular patterns (PAMPs) [37] bind to specific molecular structures called surface-expressed pattern recognition receptors (PRRs) presented by the immune cells (e.g., macrophages, dendritic cells) resident in the normal healthy tissue [33,38]. The PRRs can function as soluble proteins involved in the opsonization process, as phagocytic transmembrane receptors (e.g., the mannose receptor and dectin-1) and can be involved in complement activation [33]. To date, many classes of PRRs, including toll-like receptors (TLRs), nucleotide-binding oligomerization domain (NOD)-like receptors (NLR) and retinoic acid inducible gene-I (RIG-I)-like receptors (RLR), have been studied and characterized [38]. These receptors elicit an immune response through the activation of the transcription nuclear factor kappa B (NF-kB), which prompts the upregulation of proinflammatory cytokine genes such as IL-1 and TNF-α, responsible for the recruitment and activation of different subsets of leukocytes (e.g., neutrophils, macrophages) at the site of injury or infection [38,39]. This represents the starting point of a cascade of events responsible for changing the local environment of the surrounding tissue and vasculature [40]. The first cells to act are neutrophils, which engulf pathogens and attract other immune cells such as macrophages and other neutrophils, through the secretion of various inflammatory molecules and growth factors [41,42,43]. The infiltrating macrophages, together with the tissue-resident macrophages, adopt a proinflammatory phenotype, clearing the foreign particles/debris through phagocytosis and secreting pro-inflammatory cytokines such as IL-6, TNF-α and interferon-γ (IFN-γ), vital for the early phase of normal tissue healing and the activation of the adaptive immune system [44]. In contrast to the innate immunity, the adaptive immune system contains cells (B and T cells) capable of recognizing specific antigens, after multiple contacts, developing the so-called “immunological memory” [45]. A special subset of immune cells, which act as messengers between the innate and the adaptive immune systems, is represented by the dendritic cells. This cell population is capable of processing antigens into short peptides and presenting them on their cell surface (antigen presentation), thus exposing the antigens to the T cells for phagocytosis [46]. Another important subpopulation of cells is represented by the mastocytes, tissue-resident sentinels, capable of releasing various mediators (e.g., cytokines, chemokines) involved in leukocyte recruitment and venular permeability enhancement. Moreover, data found in the literature showed that by suppressing the activation and function of the mastocytes, an indirect interference in the mast-cell-induced recruitment of other immune cells could be observed [47].

Figure 1 offers a basic overview of the innate and adaptive immune system components that contribute to the immune reaction towards a pathogen agent/foreign body.

## 3. The Host Immune Response Following Biomaterial Implantation

Various materials (Figure 2) have been widely used in the field of regenerative medicine and tissue engineering with the purpose of restoring the lost structure and function of the injured bone tissue [48,49,50,51,52].

However, when implanted in vivo, all biomaterials can be recognized by the host as foreign bodies [29], therefore eliciting an array of cellular and tissue responses [50], which can determine the success of the osseointegration process and the biological performance of the implantable devices [53]. Moreover, when degradable biomaterials are implanted, due to their natural degradation process that takes place in a physiological environment, the immune response is additionally affected by surface changes and degradation products [28]. Furthermore, depending on the structure of the implanted biomaterial, the initial degradation can further facilitate the process, therefore leading in time to the abrupt failure of the structure. Following the in vivo implantation, host reactions include a cascade of processes that consists of blood–material interactions, provisional matrix formation, acute and chronic inflammation, development of granulation tissue, foreign body reaction and fibrous capsule development/fibrosis [54,55,56,57,58]. The presence of this cascade of events can either result in tissue remodeling and new bone formation or, in the case of a prolonged inflammatory state as a response to a foreign body reaction (FBR), it can lead to the development of the fibrous capsule and osteolysis. Nanoseconds after bio-implantation, blood from the injured vessels surrounds the implant, beginning the interaction with the biomaterial. Following this event, the blood and interstitial fluid proteins, such as albumin, fibrinogen, γ globulin, complement, vitronectin, fibronectin, sugars, lipids and ions, are spontaneously adsorbed onto the surface of the biomaterial [9,54,59]. The surface properties of the biomaterial are capable of influencing the concentrations and types of the adsorbed proteins and the further recruitment and adhesion of various cells. These characteristics are key players in the inflammatory and wound healing responses towards implantable biomaterials [60,61,62,63].Therefore, the presence of the newly acquired layer of proteins dictates the activation of the extrinsic and intrinsic coagulation systems, the complement system, platelets and immune cells and directs their interactions towards the formation of an initial thrombus at the interface between tissue and material surface, also known as the transient provisional matrix [9,22,64]. The extrinsic and intrinsic coagulation systems are activated by specific proteins such as factor XII (FXII) and tissue factor (TF). Upon its interaction with negatively charged surfaces, FXII becomes activated and starts a cascade of protein reactions, which results in thrombin release [64,65]. In turn, the released thrombin activates platelets and coagulation factors, thus enhancing the coagulation cascade at the site of the injury [66]. Moreover, thrombin transforms fibrinogen to fibrin, necessary for the formation of the primary fibrous mesh around the implanted biomaterial [28]. In addition, the rapidly adsorbed fibrinogen is involved in triggering the immune response following cloth formation and in platelet adhesion and activation [28].

The complement system represents a major host defense system, which becomes activated upon contact with the adsorbed layer of proteins formed on the surface of the implantable biomaterial [28]. Following the activation of the complement cascade at the implantation site, high concentrations of C3a and C5a are produced [67] with the purpose of triggering mast cell degranulation, increasing vascular permeability, attracting and activating granulocytes and monocytes and inducing ROS release by granulocytes [68].

The coagulation cascade and the complement system interact on the surface of the biomaterial, working together in the inflammatory cell activation. The fibrin mesh and the activated platelets that result from the coagulation cascade, together with the inflammatory mediators released during complement activation, lead to the development of the transient provisional matrix [54]. The transient provisional matrix offers biochemical, structural and cellular components for the wound healing process and FBR. Different cytokines, growth factors, mitogens and chemo-attractants, found in the provisional matrix, are capable of recruiting cells of the innate immune system at the site of the implant [54]. Moreover, the complex tridimensional structure of the fibrin network offers an adequate support system for cell migration and adhesion [54]. Hence, the provisional matrix can be seen as a natural, biodegradable support system capable of releasing active bioagents involved in the modulation of the following stages of wound healing [55]. Following the initial blood–material interplay and provisional matrix development, acute and chronic inflammatory responses follow each other. Their degree of intensity is dependent on both the extent of the injury sustained during the implantation process and on the extent of the provisional matrix formation [54]. The acute inflammatory response is characterized by the presence of neutrophils (polymorphonuclear leukocytes (PMNs)) at the implantation site, their recruitment being triggered by various chemo-attractants (TGF-β, platelet-derived growth factor (PDGF), IL-1 and leukotriene (LTB4)) released from the host-activated platelets, endothelial cells and injured tissue cells. Once recruited, they become activated and adhere to the biomaterial’s surface through a mechanism that involves integrin-driven interactions (β2 integrins) [9], triggering the phagocytic response and degranulation [69,70]. In addition, mast cells are also directly involved in the acute inflammatory response through the release of histamine and inflammation-enhancing cytokines produced during their degranulation [70]. IL-4 and IL-13, released during the degranulation process, are capable of determining the extent and degree of the subsequent development of the FBR [54]. Moreover, the adsorbed host fibrinogen, coupled with the histamine release from the mast cells, mediates the acute inflammatory response through the recruitment and adhesion of phagocytes to the implant surface [71,72]. Upon activation, the neutrophils produce and release proteolytic enzymes and ROS in an attempt to destroy the foreign bodies, the cellular debris and the pathogenic agents [73]. Moreover, they secrete significant amounts of pro-inflammatory cytokines such as IL-1β, TNF-α, IFN-γ, which will lead to the further degradation of the surrounding tissue, as well as immunoregulatory molecules such as IL-8, monocyte chemo-attractant protein (MCP)-1 and macrophage inflammatory protein (MIP)-1α [74]. The secreted chemokines are potent chemo-attractants for monocytes, macrophages, immature dendritic cells and lymphocytes [75] and their progressive increase inhibits further PMN infiltration in favor of monocyte recruitment [76]. The remaining PMNs found at the implantation site rapidly become exhausted, undergo apoptosis and are engulfed by macrophages [76], disappearing within the first two days following biomaterial implantation [54]. The acute inflammatory response normally resolves in less than one week, depending on the extent of the trauma at the implantation site, and it is followed by a chronic inflammatory phase, where monocytes/macrophages are the predominant cells involved in the inflammation evolution. The chronic inflammatory state usually lasts no longer than two weeks and is confined at the site of the implantation. However, the persistence of the acute and/or chronic inflammatory state for more than three weeks usually indicates an infection [54]. In response to the chemo-attractants and activation of cytokines released during the previous phase, the blood-circulating monocytes migrate to the implant surface, bind fibrinogen to the provisional matrix and undergo phenotypic differentiation, changing into macrophages [13]. Macrophages are key players in the wound healing process, being involved in wound debris clearance and the production of various mediators such as enzymes, cytokines, growth factors, etc. [77]. They possess high plasticity, being able to change their functional phenotype in response to stimuli received from changes in the microenvironment [29]. Therefore, they can acquire a pro-inflammatory phenotype (M1) or an anti-inflammatory phenotype (M2) [78,79]. During the early inflammatory phase, macrophages acquire a M1 phenotype, eliciting an upregulation in the pro-inflammatory mediators [80], a mechanism necessary for the normal wound healing process [81]. Adherent activated macrophages secrete chemokines [82], ROS and degrading enzymes [83,84] for further inflammatory cell recruitment and degradation of the biomaterial, respectively. Similar to the wound healing process, the adherent macrophages polarize to an M2 phenotype, characterized by a switch in their secretion profile reflected by anti-inflammatory cytokine production, reduced degradative capacity and the ability to stimulate the migration and proliferation of fibroblastic cells towards an effective bone regeneration process [77]. An ineffective switch from the M1 towards the M2 phenotype, coupled with the mechanism of frustrated phagocytosis, leads to macrophage membrane fusion and foreign body giant cell (FBGCs) formation, an event which represents a hallmark of the chronic inflammatory state. Single macrophages are able to phagocytose particles up to 5 µm in size [85], but if the particle size is larger, the cells undergo fusion to form FBGCs. The process of cell–cell fusion is fostered through the activation of basophils, mast cells and T helper (Th) cells that produce IL-4 and IL-13, cytokines that have been shown to enhance macrophage fusion on the biomaterials’ surface [86,87,88,89]. However, the mechanism through which cells of monocytic origin fuse to form multinucleated cells has not been fully characterized, and the present proposed mechanism involves three important steps. In the first step, cells need to acquire their ability to fuse; then, the fusion-competent cells migrate and attach themselves to membranes found in close proximity, and lastly, the cells undergo fusion, sharing their cellular components [90]. Moreover, in order for cells to become fusion-competent, specific fusion-inducing mediators are required, and even so, their simple presence is not sufficient if the surface of the biomaterial does not support the fusion process. Therefore, the protein layer found on the surface of the biomaterial dictates the fate of the adherent cells. In this context, a variety of proteins, such as fibrinogen, collagen, fibronectin, laminin, vitronectin, have been studied in regard to their ability to promote FBGC formation and the results showed that only vitronectin supports the fusion process [91]. Consequently, the formation of FBGCs depends heavily on the presence of the fusion-inducing stimuli and the appropriate adsorbed protein layer on the biomaterial surface.

Following the resolution of acute and chronic inflammation, the formation of the granulation tissue is recognized by the presence of macrophages and the recruitment of fibroblasts and endothelial cells at the implantation site [55]. It is thought that the granulation tissue is a precursor for fibrous capsule development, due to the fact that is separated from the implant only by a one- to two-cell layer of macrophages, monocytes and FBGCs [92]. The healing process of the tissue surrounding the implantation site, depending on the immune response, can follow two paths: bone regeneration or fibrous capsule development. The path which the implant will follow depends on the proliferative capacity of the host cells and on the extent of the provisional matrix and that of the ECM formed at the implantation site. Moreover, the synergetic action of the immune cells results in the release of various pro-fibrogenic factors, which are capable of recruiting fibroblastic and endothelial cells. Normally, the recruited fibroblasts, in an attempt to repair the injured tissue, will deposit type I and III collagen, but, in the presence of a biomaterial, excessive secretion of collagen occurs, leading to the formation of fibrotic tissue [93]. This newly formed tissue will encapsulate the biomaterial and prevent the attachment of the bone-forming cells to its surface for new bone formation. This scenario will render the biomaterial inert, failing to meet the demands of a bone substitute material, leaving the defect to be filled with fibrous tissue instead of the new bone tissue [55]. Figure 3 summarizes the key events involved in the inflammatory response to an implantable biomaterial.

## 4. The Role of the Immune System in Bone Dynamics

The immune cell involvement with the skeletal system does not only involve their role in maintaining bone homeostasis, but they also play a vital role in the bone healing process. The most recent studies in the bone tissue engineering field have reported that the bone healing process is largely dependent on the intimate relationship between cells of the immune and skeletal systems. Bone healing is a complex process that involves various time-overlapping regulations. It can be divided into four major phases (inflammatory phase, the fibrocartilaginous bone formation, bony callus formation and the bone remodeling phase), each of them involving various processes that require an intimate crosstalk between immune and bone-forming cells [29]. Among the cells of the immune system, macrophages are recognized as key players in the recovery process to reestablish the tissue integrity and function after injury [94].

### 4.1. Macrophage Plasticity and Polarization States

Macrophages are prodigious phagocytic cells, considered to be the organism’s first line of defense against infectious microorganisms and various pathogens. However, in the last few years, their indispensable role in homeostasis and the bone remodeling process has been elucidated [50].

The macrophages present one of the most diverse and adaptive transcriptomes, expressing a broad range of cell surface receptors, pro- and anti-inflammatory cytokines, growth factors, chemokines, proteolytic enzymes and many other cellular products [95,96,97]. They are derived from a distinct population of blood-circulating monocytes (CD14^hi^CD16^−^ and CD14^+^CD16^+^ monocytes) [98,99,100] that migrate and infiltrate the compromised tissue [95,101,102]. Once infiltrated, in response to the local signals associated with pathogens or traumatized tissue, the monocytes differentiate into macrophages, becoming activated and increasing their production of cytokines, chemokines or other molecules that contribute to the local microenvironment. Very much like the monocytes from which they originate from, macrophages are a heterogeneous cell population with different markers and functions [98,103,104,105,106]. They possess an extraordinary, attuned responsiveness and a diverse expression capacity which translates into high plasticity and the ability to exhibit a spectrum of polarization states, which are defined by their function and patterns of gene expression [16,106,107]. Polarized macrophages are generally referred to as having either a pro-inflammatory M1 or an anti-inflammatory M2 phenotype, similar to the Th1/Th2 nomenclature which has been used for the T helper cells [108]. The M1 phenotype is normally associated with the early stages of tissue repair due to the fact that M1 macrophages are key players in the acute inflammatory phase. The M2 cells act during the late phases of bone healing, being involved in either new bone tissue formation or the development of fibrous tissue. Both phenotypes produce and secrete different factors and cytokines that interact with osteoblastic, osteoclastic, mesenchymal and endothelial cells during the bone healing process.

The “classically activated” or the pro-inflammatory M1 phenotype emerges as result of the exposure to inflammatory signals such as IFN-γ alone or in combination with microbial products such as LPS or TNF-α [105]. M1 macrophages are recruited shortly after injury and are involved in the early immune response by enhancing the local inflammatory microenvironment through the production of pro-inflammatory cytokines and ROS as an attempt to clear pathogens or other foreign bodies from the wound site [109,110]. The M1 phenotype releases various pro-inflammatory cytokines (TNF-α, IL-1β, IL-6, IL-12, IL-23) and chemokines (α-chemokine ligands (CXCL1)-3, CXCL-5 and (CXCL8)-10), produces high levels of nitric oxide synthase (iNOS), metabolizes arginine, secretes reactive oxygen and nitric oxygen intermediates and promotes lymphocyte differentiation towards Th1 cells [106]. In the context of bone healing, the M1 phenotype dictates the initial clearance of tissue debris and induction of osteoclastogenesis, while their prolonged presence leads to a chronic inflammatory state. Regarding biomaterial implantation, the initial presence of the M1 macrophages induces the necessary inflammatory response but their prolonged presence leads to a severe FBR and fibrous encapsulation. Therefore, in order to restore the normal function of the traumatized bone tissue and avoid implant failure, a timely, favorable switch between the M1 and M2 phenotypes is necessary.

The M2 macrophages, also referred to as “alternatively activated “macrophages, are induced through exposure to various signals, such as cytokines (IL-4, IL-13 and IL-10) secreted from mast cells, basophils and other granulocytes, immunocomplexes (ICs), adenosines, glucocorticoids (GCs), arginase, TLRs and growth factors (TGF-β) [16].The M2 macrophages release anti-inflammatory cytokines such as IL-10, express high levels of scavenging molecules and mannose and galactose receptors (CD163, CD206, CCR2) [111], produce arginase-1 (Arg-1), which is a substrate of iNOS, and are involved in the polarized Th2 reactions. In addition, the M2 macrophage population is heterogenous, encompassing a range of different subsets, namely M2a, M2b and M2c [95], each with its own distinct inductors, markers and functions. The M2a subset is normally induced as a result of either IL-4 or IL-13 stimulation and its main role is to support the wound healing process by secreting high levels of Arg-1, which contributes to the production of collagen and fibroblast-stimulating factors [112]. The M2b subpopulation is induced by either Arg-1, ICs or TLR agonists and is involved in suppressing the inflammatory reaction by Il-10 production [113]. M2c macrophages are induced by IL-10 and play a key role in the tissue remodeling phase by releasing IL-10 and TGF-β [114]. In the context of biomaterial implantation, the presence of the anti-inflammatory cytokines and tissue remodeling response leads to neovascularization and inhibition of fibrotic encapsulation, which improves the implant biological function and integration.

However, this classification of macrophages into two different phenotypes represents a simplistic picture of the in vivo situation; therefore, this concept is slowly being replaced by the idea of a continuum of different activated states according to the temporal presentation of stimuli that the macrophages are exposed to. This suggests that the presence of different macrophage phenotypes in the same microenvironment can be used as a potential strategy in order to obtain contrastive remodeling mechanisms for a reduced inflammatory reaction. Figure 4 shows the inducers and the released molecules for each macrophage subtype.

### 4.2. The Crosstalk between Immune and Bone-Forming Cells

Besides their involvement in removing tissue debris and confining and reducing the spread of the inflammation, current research is shedding light on their involvement in processes such as osteogenesis and osteoclastogenesis. The interplays between the immune and bone-forming cells are multiple and follow different pathways. For example, a series of studies demonstrated that the secretion of cytokines and chemo-attractants by the immune system cells attracts mesenchymal stem cells (MSCs) to the injury site and modulates the osteogenic and osteoclastogenic processes [115,116,117]. On the other hand, it is known that osteal macrophages (OsteoMacs) are involved in hard callus maturing [46]. Chang et al. [118] reported that the lack of macrophages inhibited osteoblast-mediated bone formation in vivo. Altogether, various cytokines, signaling molecules, transcription factors and receptors can induce a favorable crosstalk between immune and skeletal cells, due to their common origins.

#### 4.2.1. The Immune Response and the Osteogenic Process

Osteogenesis is the bone formation process in which early bone is developed and ECM is mineralized [29]. The immune cells are closely interconnected with the development of new bone tissue, playing an indispensable regulatory role in the osteogenic process. Similar to other inflammatory responses, during the bone healing process, macrophages play a vital role in bone regeneration, by influencing the local microenvironment through the secretion of various cytokines. Recently, it has been reported that both pro- and anti-inflammatory macrophages can have a positive effect on the new bone formation, therefore playing a major role in controlling osteogenesis-related processes. The secreted cytokines can regulate the inflammatory response, controlling the differentiation of MSCs into bone forming cells and, once matured, their function.

The M1 phenotype releases inflammatory cytokines such as TNF-α, IL-6, IL-1β and IL-23 [29]. It was reported that TNF-α could induce the upregulation of alkaline phosphatase (ALP) activity, an early osteogenic marker, and stimulate the mineralization process by MSCs, through the activation of the NF-kB signaling pathway [119]. Moreover, by pre-treating LPS-stimulated growth medium with a TNF-α-neutralizing antibody, the stimulatory effect on the ALP activity was attenuated [120]. In addition, the vital role of the pro-inflammatory cytokine IL-6 in the early stages of fracture healing has been reported by Yang et al. [121] in their study, where IL-6 knockout mice were found to present a delay in callus maturity, mineralization and tissue remodeling. On the other hand, the knockout of Oncostatin M (OSM), a cytokine that pertains to the IL-6 family, in the early stages of fracture healing led to a reduced amount of new bone [122]. Furthermore, delivering pro-inflammatory cytokines such as TNF-α, IL-17 and INF-γ, in controlled dosages, has been shown to activate the autologous differentiation factor from MSCs, inducing their differentiation into mature osteoblasts [120]. However, due to their bimodal role, in the specialized literature, their inhibitory effects on the osteogenic process have been reported. For instance, TNF-α can suppress the differentiation of the osteoblastic cells by reducing the levels of bone morphogenetic protein-2 (BMP-2) and enhancing the expression of canonical Wnt signaling pathway inhibitors dickkopf-1 (DKK-1) and sclerostin (SOST). Furthermore, it can stimulate the apoptotic process of the osteoblastic cells [123,124,125]. In addition, high levels of TNF-α and INF-γ released by the proinflammatory T cells can suppress bone marrow mesenchymal stem cells’ (BMMSCs’) ability to mediate the bone regeneration process through a downregulation in the run-related transcription factor 2 (Runx-2) pathway. However, this inhibitory effect can be overcome by the administration of anti-inflammatory drugs such as aspirin [126]. It is hypothesized that anti-inflammatory drugs lead to the activation of the transcription factor NF-kB, which in turn enhances the degradation of an important component (β-catenin) of the Wnt osteogenic signaling pathway [127]. These observations suggests that the effects of the pro-inflammatory mediators on bone dynamics are time- and dose-dependent. Therefore, the proper timing and concentration of these mediators is of importance in order to elicit new bone formation and avoid bone resorption.

Opposing the M1 phenotype, the M2 pro-healing macrophages play an important role during the middle and late stages of fracture healing, and depending on the release profile of the inflammatory cytokines, they can induce the formation of either new bone or fibrous tissue. The M2 phenotype secretes anti-inflammatory cytokines such as IL-10, TGF-β, IL-1ra and pro-osteogenic molecules such as BMP-2 and VEGF [128]. IL-10 was shown to enhance the osteoblastic differentiation in an IL-10-depleted mouse model [129]. Moreover, by stimulating human MSCs with TGF-β, their differentiation into osteoblasts through autocrine BMP signaling was reported [130]. IL-1ra acts as an inhibitor for IL-1, regulating its adverse effects. However, it was reported that the prolonged release of factors such as TNF-α, TGF-β1 and TGF-β3 leads to the formation of scar tissue and a delay in the wound healing process [131].

This close relationship between the immune cells and the bone regeneration process demonstrates that the traditional strategy of focusing on the interaction between the biomaterial and bone forming cells is not sufficient, as they do not reflect the in vivo situation, in which the immune cells play a role during the process of wound repair. Therefore, the observation of the intimate interplay between immune and skeletal systems represents a strong argument in considering the importance of the immune response in designing new bone substitutes.

#### 4.2.2. The Immune Response and the Osteoclastogenic Process

Bone regeneration is a complex process in which its two stages of bone formation and resorption are balanced. Bone resorption is a two-step process that starts with the proliferation and differentiation of osteoclast precursors and continues with the degradation of the organic and inorganic phases of the bone tissue [132]. The osteoclasts are vital players in the bone healing cascade, being involved in the ECM resorption and depletion of the tissue area prior to new bone deposition. The osteoclast maturation occurs through a complex receptor activator system that involves the interaction between macrophages and bone-forming cells [133]. In addition to osteoclastic differentiation, the immune system is tightly connected to the osteoclastogenic process through the release of important cytokines and chemokines from macrophages. The immune system modulates the osteoclastogenic process through the involvement of three main cytokines—M-CSF, receptor activator of NF-kB ligand (RANKL) and osteoprotegerin (OPG) [13]. During bone remodeling, under M-CSF and RANKL stimulation, macrophages differentiate into osteoclasts. RANKL binds itself to a receptor found on the surface of the osteoclast precursors, RANK, leading to an upregulation of the gene expression for the survival and differentiation of osteoclasts, through the TNF receptor-associated factor 6 (TRAF6), NF-kB, activator protein-1 (AP-1) and nuclear factor of activated T cells 2 (NFAT2) [15]. Similarly, M-CSF binds to its associated receptor c-FMS, found on the surface of the osteoclast precursors, leading to osteoclast differentiation through the Akt and MAP kinase pathways [134]. Moreover, inflammatory cytokines such as IL-6 and TNF-α play an important role in the RANKL/RANK/OPG systems and, implicitly, in the osteoclastogenic process. IL-6 is known to induce the expression of RANKL [155/135], while TNF-α indirectly modulates the expression of RANKL by stromal cells through IL-1 stimulation. TNF-α was also demonstrated to inhibit OPG expression and stimulate M-CFS production in bone-forming cells [135,136]. Furthermore, through the stimulation of the apoptotic process of osteocytes, TNF-α is capable of attracting osteoclasts [137]. Similarly to TNF-α, IL-17, IL-23 and IL-1 were capable of inducing a positive effect on the osteoclastogenic process through the enhancement of RANKL expression (IL-17 and IL-23) and stimulation of MCS-F production (IL-1) in bone-forming cells [138,139,140]. In contrast, some cytokines can elicit an inhibitory effect on the osteoclastogenic process. For example, IL-10 downregulates the expression of the nuclear factor of activated T cells, cytoplasmatic 1 (NFATc1), necessary for osteoclast differentiation, therefore inhibiting the resorptive process. Moreover, it was reported that in a mixed osteoblast–osteoclast cell culture, IL-12 was capable of inhibiting RANKL-mediated osteoclastogenesis and TNF-α stimulated osteoclast differentiation [141,142]. OPG, a decoy receptor for RANKL, can inhibit both the differentiation and function of osteoclasts, by binding itself to the RANKL receptor and disrupting the RANKL/RANK interaction [133,143].

Apart from macrophages, other immune cells actively participate in the osteoclastogenic process. For instance, RANKL is expressed not only by osteoblastic cells but also by the activated T cells and neutrophils, indicating their involvement in the osteoclastogenic process [144,145]. Moreover, a reduction in the number of mast cells led to a suppression in the bone remodeling process, whereas an increase in the systemic mastocytes led to an enhancement in the amount of bone loss [146,147].

The close relationship between the immune cells and osteoclasts plays an important role in many bone pathologies, such as rheumatoid arthritis, osteoarthritis, etc. The presence of a prolonged pro-inflammatory state leads to an increase in the RANKL/OPG ratio and enhanced osteoclast activity [148]. This has, as a direct consequence, a shift in the bone remodeling process towards an accelerated bone resorption process, characterized by a derangement in the organic and mineral components, which, in the end, will result in excessive bone loss.

## 5. Development of Bone Biomaterials with Immunomodulatory Properties

The implantation of a bone biomaterial triggers multiple directional immune responses, resulting from the damage caused to the host tissue and from the interplay between the biomaterial and the surrounding microenvironment. When facing the host immune system, the implantable biomaterial is not simply a passive target, but elicits significant effects that modulate the extent and type of implant-mediated immune response. Therefore, understanding the influence of the biomaterial properties on different stages of bone healing could lead to favorable results for the osseointegration process. The ideal bone substitute should be capable of stimulating favorable crosstalk between the cells of the immune and skeletal systems in the different stages of bone healing. The fate of the implant rests heavily on the immune response elicited by the biomaterial; therefore, the need for materials capable of instructing the immune system to elicit an adequate immune response has become apparent. The concept of biomaterial-associated osteoimmunomodulation highlights two main developing strategies: the need for a proper evaluation of the biomaterial’s effect on the immune responses and the development of biomaterials capable of modulating a proper immune reaction at the implantation site [149,150]. In this context, a series of strategies, such as modification of the chemical/topographical characteristics or the incorporation of bioactive molecules (Table 1), have been proposed with the purpose of designing biomaterials capable of controlling macrophage polarization and the positive crosstalk with the bone-forming cells. The following section synthesizes the recent advancements in the field of immunomodulatory biomaterials for osteogenesis and osteoclastogenesis, with a special focus on the tunable properties of the bone biomaterials.

**Table 1 materials-14-01357-t001:** Surface properties that influence the immune responses towards an implantable biomaterial.

Tunable Properties		Effect of Immune Cells	Ref.
Surface chemistry	Wettability	hydrophobicity: ↑ monocyte adhesion hydrophilicity: ↓ macrophage adhesion;	[53,82,151,152,153,154,155,156,157,158,159,160]
	Charge	anionic/neutral particles: ↓ inflammatory reaction cationic species: ↑ inflammation;	[30,160,161,162,163]
Surface topography	Roughness	induces significant immune reactions, influences immune cell adhesion;	[30,158,160,164,165,166,167,168,169,170,171,172,173,174]
	Particle size	influences the immune reaction, no consensus has been reached on size;	[161,175,176,177]
	Porosity/pore size	larger pore size: ↓ inflammation, ↑ angiogenic process;	[178,179,180,181,182]
Delivery of biological molecules		elicit immunoregulatory effects	[2,183,184,185,186,187,188,189,190,191,192,193,194,195,196,197]

Data are collected from the specialized literature as seen in the flowchart from Appendix A. ↑ indicates enhancement; **↓** indicates inhibition.

### 5.1. Immunomodulatory Biomaterials for Osteogenesis

#### 5.1.1. Surface Chemistry Alterations

The surface chemistry of the biomaterial is an important aspect that determines its interaction with the biological microenvironment, in terms of protein adsorption and cellular responses. As previously described, the biomaterial surface’s interaction with the adsorbed layer of proteins is vital for the appearance of the immune response towards the implant [198,199]. In this context, a series of studies reported that the immune cell response can be influenced by altering different surface chemical characteristics, such as wettability [152], surface charge [30] or functional groups [152].

The biomaterial’s wettability is strongly associated with the protein layer adsorption, blood clot formation and fibrin formation. Highly hydrophilic biomaterials are normally protein-resistant [2] while hydrophobic biomaterials present an intrinsic immunogenicity [154,155] since the host immune system can recognize the hydrophobic portions of different biological molecules as DAMPs, leading to PRR activation and molecule elimination [200]. In this context, the biomaterial’s hydrophobicity or hydrophilicity represents a crucial factor which influences the protein adsorption. Visalakshan et al. [53] evaluated the role of surface chemistry and wettability in serum-derived protein layer formation on the surface of a biomaterial and the subsequent effects on the elicited immune response. In this study, a substrate-independent technique (plasma polymerization) was used to obtain nano-thin biomaterial coatings with different chemical functionalities and a spectrum of surface charges and levels of wettability. The results showed that the type and amount of the adsorbed proteins was significantly influenced by surface chemistry and wettability. Therefore, the hydrophilic carboxyl surfaces favored albumin adsorption, while the hydrophobic hydrocarbon surfaces favored IgG2 adsorption. Furthermore, Thevenot et al. [154] investigated the effect of gold nanoparticles functionalized with increasing hydrophobic chemical groups on immune cells isolated from mice spleen and the results showed that particles with increased hydrophobicity stimulated the gene expression profile of TNF-α and IFN-γ. Likewise, Kakizawa et al. [155] designed monodisperse silica nanoparticles coated with different poly-(amino acids) of various degrees of hydrophobicity and reported that the secretion of IL-1β and IFN-γ is correlated with the hydrophobicity of the poly-(amino acids). Furthermore, it was demonstrated that biomaterials with increased hydrophilicity enhanced the bone regeneration process. Li et al. [156] demonstrated that pristine titanium (Ti) surfaces with lower hydrophilicity elicited an enhanced secretion of several pro-inflammatory cytokines such as TNF-α, MCP-1 and IL-1β, in comparison to Ti surfaces functionalized with heparin/fibronectin. On the other hand, Alfarsi et al. [157] reported that surface hydrophilic modifications downregulated the gene expression profiles of several pro-inflammatory cytokines and of their associated proteins. Hamlet et al. [158] investigated the comparative effect of two hydrophilic modified sandblasted/acid-etched (modSLA) and SLA Ti surfaces and the results showed a reduction in the gene expression profiles of the TNF-α, IL-1β, IL-1α cytokines and MCP-1 chemokine from the macrophages seeded on the surface of the hydrophilic Ti. In another study, Dai et al. [159] evaluated the effect of hydrophilic Ti disks on RAW 264.7 murine macrophages and the results obtained revealed that the hydrophilic surface was able to upregulate the secretion of IL-10 and downregulate the secretion TNF-α, respectively. Moreover, Hotchkiss et al. [160] investigated the effect of an oxygen-plasma-generated, hydrophilic Ti disk on primary murine macrophages isolated from C57BL/6 mice and the results showed that the hydrophilic Ti surface could downregulate the levels of pro-inflammatory cytokines and upregulate the levels of anti-inflammatory cytokines. Furthermore, to combat the immunogenic effects of the hydrophobic surfaces, hydrophilic molecules such as polyethylene glycol (PEG) and polyethylene oxide (PEO) were added as monolayer coatings to delivery platforms and tissue engineering constructs in order to increase their hydrophilicity and reduce protein adsorption [201,202]. However, increased hydrophilicity results in high protein adsorption resistance, which can lead to decreased interactions with the immune cells, which in turn may reduce the immunomodulatory effects [201,202,203]. Even though this reduction in the interplay with the immune cells could be favorable in counteracting undesirable pro-inflammatory responses, future strategies could leverage changes in surface chemistry to modulate the immune reaction towards natural healing responses to injury. Another important surface aspect is represented by the chemical groups. The surface charge of the biomaterial plays a crucial part in immune response modulation [162,163,204,205]. Therefore, various functional groups, such as amino (-NH2), hydroxyl (-OH), carboxyl (-COOH), are commonly studied. A series of in vivo studies demonstrated that the amino and hydroxyl groups could induce the enhanced infiltration of immune cells [206,207,208] and the development of a thicker fibrotic capsule surrounding the implant [207,209]. In addition, cell differentiation and focal adhesions [10,210] were enhanced to different degrees by the presence of hydroxyl groups, followed by amino and carboxyl ones [10]. Bartneck et al. [162] reported that, depending on the exposed functional groups, surface-charge-modified nanorods were capable of altering the inflammatory profiles of macrophages. The amino-terminated nanorods exhibited a positive surface charge, therefore inducing an anti-inflammatory M2 phenotype. On the contrary, the carboxyl-terminated nanorods induced a switch towards the M1 pro-inflammatory phenotype, due to the negative surface charge. However, a series of studies have reported that positively charged particles can lead to enhanced activation of the inflammasome in comparison to negatively charged particles [163]. Moreover, other works showed that the immune function can be inhibited or blocked by negatively charged particles [205,211], due to changes in the migratory behavior and function of macrophages. These actions can disrupt the macrophage-mediated inflammatory response and promote regulatory T cell phenotypes [211]. For example, Getts et al. [211] synthesized negatively charged particles from carboxylate poly(lactide-co-glycolide) (PLGA), polystyrene and microdiamonds and reported that all the tested particles could suppress the inflammatory activity of macrophages.

In addition, several bone biomaterials are not inert but bioactive; therefore, their surface is in dynamic evolution due to the release or uptake of soluble products when in contact with a physiological environment. This process is even more complex in the case of resorbable biomaterials because the interaction with the surrounding cells will not only take place through direct contact with the biomaterial surface but also through the products (e.g., ions) released in the local microenvironment. Therefore, the effect of the chemical composition on the immune cells’ response is an important factor that should be taken into consideration. For instance, Chen et al. [212] used a chemical immersion method to coat Mg scaffolds with β-tricalcium phosphate (β-TCP) as a strategy to manipulate their osteoimmunomodulatory properties, and the outcome suggested that the coating could induce a proper and effective switch towards the M2 macrophage phenotype, therefore leading to enhanced bone marrow-derived stem cell (BMSC) differentiation and inhibition of the inflammatory state and osteoclastogenic process. In another study, Chen et al. [213] investigated the effect of Mg-β-TCP extracts on the macrophage response and the results showed that, in response to the calcium-sensing receptor (CaSR) pathway activation, the macrophage polarized towards the anti-inflammatory M2 phenotype. Moreover, the expression of BMP-2 was significantly upregulated by β-TCP stimulation. In terms of BMSCs’ differentiation, their osteogenic differentiation was significantly enhanced, therefore suggesting the important role of macrophages in biomaterial-mediated osteogenesis. Similarly, Chen at el. [214] demonstrated that a biphasic calcium phosphate (BCP) scaffold consisting of β-TCP:HA (80:20) was able to generate a more favorable osteoimmunomodulatory microenvironment in comparison to β-TCP and HA (hydroxyapatite) alone, both in vitro and in vivo. The BCP scaffold led to an enhancement in the number of pro-healing CD206+ M2 macrophages and the induction of the new bone-forming process in a murine model. Wang et al. [215] reported that another BCP with a similar composition β-TCP:HA (70:30) could elicit the early expression of various growth factors from the RAW 264.7 cell line and enhance the production of osteocalcin (OCN) and ECM mineralization by MSCs when cultured with condition media collected from the seeded murine macrophages. Moreover, following implantation in a physiological environment, the bioactive bone biomaterials undergo a degradation process which results in the corrosion products’ release [216]. Amongst the corrosion products, ions can elicit significant effects on the local biological microenvironment [217,218,219], and a summary can be found in Table 2.

**Table 2 materials-14-01357-t002:** The effects of various ions on the immune response and bone-related events.

Ion	Effect on the Immune Response and Bone Events		Ref.
Calcium (Ca)	involved in the noncanonical Wn5A/Ca2+ signaling pathway and the CaSr signaling cascade		[220,221,222,223]
Magnesium (Mg)	In Vitro	↓ pro-inflammatory cytokine production through the inhibition of the toll- like receptor (TLR) pathway	[224]
		↓ expression levels of TNF-α and IL-6 ↑ production of TGF-β1 in macrophages	[225]
	In vivo	↓ osteoclastogenic process ↑ osteogenic cell recruitment	[226]
Silicon (Si)	In vivo	↑ local inflammatory response	[227]
	In Vitro	contradictory results have been reported, proving the ion’s inertness	[228]
Zinc (Zn)	In Vitro	↑ anti-inflammatory cytokine production (IL-10) ↓TNF-αand IL-1β secretion, through TLR-4 pathway modulation	[229,230,231,232]
		ZnO NP: ↑ osteogenic process of osteoblasts	[233]
		↓ inflammatory activity of RAW264.7 cells ↓differentiation and formation of mature osteoclasts	[234]
Cobalt (Co)	In Vitro:	↑pro-inflammatory effects via the hypoxia-inducible factors (HIFs)	[235,236]
Strontium (Sr)	In Vitro	↓ TNF-α production in human primary monocytes	[237,238]

↑ indicates enhancement; **↓** indicates inhibition.

Based on these observations, the strategy to modulate the immune response by the controlled release of a defined combination of bioactive elements can be considered a worthy approach.

Altogether, the presented results offer important information regarding the altering of the biomaterials’ characteristics in order to obtain the desired cellular biological behavior. However, future studies are still necessary to fully comprehend the complex relationship between biomaterials’ surface chemistry and immune cell response.

#### 5.1.2. Physical Property Alterations

Implantable devices display inherited physical characteristics, either introduced or resulting from the manufacturing process [11]; therefore, in addition to the surface chemistry aspects, the biomaterial’s topography, roughness, porosity and pore size can influence the immune cells’ plasticity [239,240,241,242,243,244,245,246,247,248,249,250], function and interaction with bone-forming cells [251,252]. By modulating the physical surface properties of the biomaterials, modifications in the adsorbed protein layer and signaling transduction can occur, therefore leading to changes in the cellular behavior [2]. In the last few decades, the surface modification strategy has attracted more and more attention in the field of implantable devices, but only recently have the osteoimmunomodulatory properties of the surface been taken into consideration as a potential method to regulate the local microenvironment for bone tissue engineering [107].

An extensively studied modification method is represented by the surface roughness, a biomaterial characteristic which can influence the interaction with the immune cells [164,165,253,254,255]. For instance, the roughness of Ti has been reported to influence immune cells’ attachment and spreading [165] and modulate the production and secretion of pro-inflammatory cytokines and chemokines [164]. Li et al. [256] investigated the influence of a micro-arc oxidation (MAO)-modified Ti surface on the inflammatory microenvironment and the results suggested that the modified surface was capable of generating a favorable inflammatory microenvironment by controlling the inflammatory mediators’ production at both stages following implantation (before and after osteoblast recruitment to the surface of the biomaterial). In general, roughness can be presented on a microscale, and there has been evidence that micropatterned surfaces can elicit beneficial effects on the osteoimmune microenvironment, leading to an improvement in the implantation success rate [2,13]. For example, Vlacic-Zischke et al. [166] developed microroughness on the surface of a Ti substrate via a sandblasting acid-etching method and the results obtained revealed that the modified surface increased the level of TGF-β signaling and stimulated osteoblast differentiation. Furthermore, Hotchkiss et al. [160] reported that Ti surfaces altered with microroughness promoted the M2 macrophage phenotype switch and an increase in IL-4 and IL-10 cytokine production, in comparison with the smooth Ti substrate, which promoted M1 polarization. In another study, Zhu et al. [167] compared a smooth-surface BCP with a BCP with a micro-whisker and nanoparticle hybrid-structured surface (hBCP) and it was observed that the hBCP substrate could downregulate the expression levels of the TNF-α and IL-6 pro-inflammatory cytokines. Uddin et al. [168] used a synergistic surface modification combining deep ball burnishing and HA coating for a commercial AZ31 Mg alloy and the results indicated that the burnished surface reduced pro-inflammatory cytokine production and increased anti-inflammatory cytokine release, in comparison to the untreated support. In addition, due to the increased roughness of the modified surface, an improvement in the coating adhesion strength could be observed. Moreover, the microroughness of the surface does not only influence the cytokine production but the angiogenic process and BMSCs’ function. Yang et al. [169] showed that the proliferation and recruitment of rat BMSCs can be promoted by the interactions between the rough surface of the Ti substrate and circulating blood. These observations suggested that microroughness could be beneficial for the healing and bone regeneration processes. However, Hamlet et al. [158] reported contradictory results, demonstrating that Ti substrates with modified microroughness promoted an enhanced pro-inflammatory cytokine profile.

Since the surface roughness of the natural bone tissue is estimated to be around 32 nm, recently, nanoscaled biomaterials have been extensively studied. Biomaterials with surface modulation at a nanoscale level can directly influence important processes such as cell adhesion and proliferation and modulate osteogenic events [257]. Chen et al. [30] developed plasma-polymerized allyalamine surfaces in which gold nanoparticles of different size (16, 38 and 68 nm) were immobilized to modulate the immune cell response. The results obtained suggested that the scale of the nanotopography was able to significantly modulate the immune microenvironment with changes in the gene expression profile of the inflammatory cytokines, osteoclastic activities and osteogenic and angiogenic factors. Furthermore, it was observed that the 68-nm surface topography elicited the most promising outcome in terms of osteogenic differentiation of the BMSCs. Similarly, Dalby et al. [170] demonstrated that nanostructures are capable of stimulating human MSCs to produce and secrete bone minerals even in the absence of special osteogenic agents. In terms of Ti-based biomaterials, a simple and easy method to modify the surface topography is the employment of the electrochemical anodization method [258], which leads to the formation of self-ordered TiO_2_ nanostructures such as nanofibers, nanotubes, nanorods, nanoarrays, nanowires and nanosheets. Neacsu et al. [171] investigated the influence of TiO_2_ nanotubes with a diameter of 78 nm on the in vitro behavior of RAW 264.7 cells under both standard and pro-inflammatory conditions and the results obtained suggested that the nanostructured surface significantly reduced the release of inflammatory mediators and induction of FBGCs, as compared to the commercial pure Ti surface. The same group conducted a more in-depth study regarding the mechanism through which the developed nanotube-modified surface attenuates the inflammatory response of macrophages [172]. It was shown that the nanotubular surface reduced the LPS-induced phosphorylation of mitogen-activated protein kinase (MAPK), IkB-α, IKKβ and inhibited the nuclear translocation of NF-kB-p65. These findings, coupled with the inhibition of NO release and MCP-1 production, suggested that the nanotubular TiO_2_ surfaces can suppress the inflammatory activity of macrophages through the inhibition of the MAPK and nuclear factor kappa-light-chain-enhancer of activated B cells (NF-kB) pathways. Ma et al. [173] developed TiO_2_-modified surfaces with different roughness (6–12 nm) and the in vivo results indicated the enhanced secretion of the pro-inflammatory cytokines associated with higher roughness. Another study [174] investigated the influence of two different dimensions of TiO_2_ nanotubes fabricated via the anodic oxidation method at 10 V (NT 10) and 20 V (NT20) on the macrophage behavior and the generated osteoimmunomodulatory microenvironment. The obtained data showed that the nanotubular TiO_2_ surfaces could modulate the macrophage polarization state, with the larger nanotubular surface (NT 20) showing the smaller M1/M2 ratio and enhanced expression of IL-10 and arginase-1 (Arg-1). On the other hand, the NT 10 surface promoted the differentiation of macrophages towards the M1 phenotype and the production and release of high levels of IL-1β, TNF-α and iNOS. In addition, decorating the Ti surfaces with nanoscale coatings can offer advanced features to the biomaterial that enhance the osseointegration process [259]. For instance, Bai et al. [260] developed a microporous TiO_2_ coating doped with HA nanoparticles on the surface of pure Ti using the MAO method and different annealing temperatures. The in vitro results suggested that the MAO-650 surface did not only support the proliferation and differentiation processes of the bone-forming cells but also elicited a favorable osteoimmunomodulatory effect by inhibiting the inflammatory activity of macrophages. In another study, Qiao et al. [261] deposited an Mg-incorporated NT array (MgN) coating on the surface of Ti and the in vitro and in vivo results showed that the newly developed coating may have endowed the surface with immunomodulating features by eliciting an inhibitory effect on the inflammatory response of macrophages. Unlike random roughness, various microfabrication methods have been employed to obtain patterns with a desired shape and size on the biomaterial surface [173]. In this context, numerous studies investigated the cells’ ability to recognize specific shapes and the results showed that the cellular response can be pattern-dependent. For instance, McWhorther et al. [176] reported that cell morphology dictates the macrophage phenotype, with an elongated morphology promoting a switch towards an M2 macrophage subtype and enhanced expression of the anti-inflammatory markers. This observation was supported by another study [107] with similar results. Here, elongated macrophages seeded on striped patterns promoted a switch towards the M2 macrophage phenotype. Furthermore, Luu et al. [177] developed Ti substrates with 400–500-nm-wide grooves and observed that the macrophage elongation occurred along the direction of the patterns. In addition, the macrophages presented an anti-inflammatory gene expression tendency, with higher levels of IL-10 secretion and decreased production of TNF-α.

With a morphological structure similar to that of natural collagen fibrils [262], electrospun nanofibers are regarded as potential bone regeneration constructs. Saion et al. [263] investigated the effect of various fibrous poly (L-lactic)-acid (PLLA) scaffolds with different alignments and fiber diameters. The results obtained suggested that the secretion of the pro-inflammatory cytokines is dependent on the fiber diameter. Therefore, in the case of the PLLA films, a higher level of cell infiltration and FBGC formation could be observed, whereas the nanofibrous PLLA scaffold induced a reduced inflammatory reaction compared to both microfibrous and simple films. However, despite the large body of information regarding the effect of various topographical features on cell behavior, future investigations are necessary since the role of osteoimmunomodulation is not fully comprehended.

Since the infiltration of oxygen and nutrients may determine the fate and polarization of macrophages, the porosity and pore size of the biomaterial have been acknowledged as another relevant surface characteristic [182,261]. Smaller pores could disrupt the nutrients and oxygen diffusion, especially in the center of the implantable device, thus resulting in a local hypoxic microenvironment [178]. In turn, the local hypoxic environment may enhance the local inflammatory reaction, leading to the formation of the granulation tissue and the complete blockage of the small pores, thereby creating a barrier between the surrounding bone cells and the implant. This path results in impaired bone tissue regeneration and implant failure [97]. Moreover, a proper hypoxic environment can stimulate the release of the angiogenic growth factors necessary for the formation of new blood vessels from the local host tissue [178]. Therefore, the surface of the biomaterials should present an appropriate pore size capable of inducing a moderate hypoxia environment, which can hinder the inflammatory reaction but promote the angiogenic effects. Klinge et al. [179] reported that pores with a size range of 90–120 µm lead to chondrogenesis and reduce the vascularization process, while larger pores with a diameter of 350 µm promote the osteogenic and angiogenic processes. In addition to its relevance in the bone-forming cell behavior, the pore size can influence the host immune system and its interaction with the implantable device [264]. It was reported that by increasing the pore size, a reduction in the activity of the FBR could be observed [179,180]. However, the underlying mechanism is still not fully elucidated, but it has been proposed that it may be related to the polarization of the macrophages [181,265,266] due to the fact that a correlation between the increasing fiber/pore size and the upregulation of the M2 markers could be observed [181]. For instance, Garg et al. [181] reported that an increase in the pore size and porosity of a polydioxanone scaffold led to an enhancement in M2 macrophage markers. Furthermore, surfaces with larger pores downregulated iNOS production, in comparison to smaller pores, which promoted a switch towards the M1 phenotype. Similarly, Chen et al. [182] demonstrated that surfaces with pores in the size range of 100–200 nm presented cells with a round-shaped morphology and increased expression of M2 phenotype markers. Sussman et al. [267] used poly(2-hydroxyethyl methacrylate) (pHEMA) and poly(methyl methacrylate) (PMMA) to evaluate the effect of the microsized pores on macrophage polarization and the results showed that surfaces with a pore size of 34 µm promoted the expression of M1 phenotype markers (iNOS, IL-1R1) upon host implantation.

Finally, the balance between the biomaterial porosity and the structural robustness must be also considered in order to ensure that its strength is not compromised [268]. Importantly, increased porosity can influence macrophage function and the regenerative microenvironment, whereas changes in the biomaterial’s structure may have a negative impact on its mechanical strength. In the case of implantable devices designed to replace tissues with structural functions such as bone, where the mechanical strength is absolutely necessary, this particular aspect is extremely important. Therefore, even if the pore size of the biomaterial can be tuned to promote a favorable switch in the macrophage polarization state, a deeper understanding of the interaction between these immunological outcomes and the material properties is necessary [10,269]. Moreover, the reported results are often contradictory and difficult to correlate and compare due to the variations amongst the surface topographies. This observation highlights the importance of using the proper cell types for a given implant purpose to identify the optimal properties capable of promoting the desired in vivo response.

#### 5.1.3. Delivery of Cytokines and Biological Molecules

Beyond chemistry and topography modifications, the incorporation of different bioactive molecules has been widely employed as a strategy to modulate the crosstalk between the osteoblast and immune cells. The osteoimmune microenvironment presents various cytokines and signaling factors involved in multiple signaling pathways, some of which have not been fully elucidated. Therefore, the selective process of the bioactive molecules should be done with caution. In order to elicit a positive inflammatory response, various active molecules, such as inflammatory cytokines [183,184,185], growth factors [186] or extracellular matrix components [187,188], have been incorporated into different biomaterials.

Due to the fact that the inflammatory response is the starting point of the healing process, the use of pro- and anti-inflammatory molecules has been widely investigated as an approach for the polarization of the immune cells. For example, Kara et al. [2] evaluated the potential of a newly designed scaffold system to sequentially deliver a short release of IFN-γ and a sustained release of IL-4 to macrophages, in order to obtain a polarization of M1 and M2 macrophage phenotypes, respectively. The results showed that the developed scaffolds were capable of modulating both the osteoimmune microenvironment by controlling the cytokine secretion profiles and the angiogenic behaviors. Similarly, Spiller et al. [189] developed decellularized bone scaffolds capable of sequential cytokine release, and the results indicated that the synergistic action of the M1 and M2 phenotypes led to an enhancement in the osteogenic process due to the increased growth factor secretion from macrophages. By immobilizing on the surface of a self-assembled monolayer made of Cr and Au, a fusion protein of recombinant human IL-1 receptor antagonist and elastin-like peptide (IL-1ra-ELP) prepared through the transformation of *Escherichia coli*, Kim et al. [190] observed an attenuated pro-inflammatory cytokine profile favorable to the osteogenic process. Furthermore, Li et al. [191] investigated the effect of the sequential polarization of macrophages through the incorporation of IFN-γ into CaSiO_3_/β-TCP scaffolds implanted subcutaneously in a mice model, and the outcomes suggested that, whilst the pro-inflammatory molecule elicited the M1 phenotype in the short term, the continuous release of Si ions from the scaffold determined the M2 phenotype switch. In another study, Alhamdi et al. [192] evaluated the in vitro effect of a biomimetic CaP coating functionalized with simvastatin and IFN-γ and the results indicated a sequential polarization of human monocyte line, THP-1. Recently, Croes et al. [193] have incorporated IL-17 into a β-TCP/HA scaffold and the results showed that the construct with IL-17 included was capable of stimulating the ingrowth of the vascularized connective tissue in a rabbit model, therefore proving its osteogenic potential. Overall, the introduction and delivery of pro- and anti-inflammatory cytokines into implantable biomaterials may prevent undesired side effects and stimulate M2 phenotype polarization.

In addition, the delivery of proteins such as BMP-2 represents a widely investigated method to stimulate the osteogenic process. Wei et al. [194] investigated the immunoregulatory role of a gelatin sponge functionalized with 20 mg/mL BMP-2 on the macrophage behavior and osteogenic process. The results showed that BMP-2 delivery led to increased macrophage recruitment and passive control of the osteogenic process through immunosuppression due to the reduction of M1 phenotype markers such as IL-1β, IL-6 and iNOS. Therefore, these findings suggest that under inflammatory conditions, BMP-2 can induce positive immunoregulatory modulation. Due to its wide range of organic molecules, such as collagen, growth factors, enzymes and glycosaminoglycans (GAGs), capable of modulating the cellular behaviors, the ECM has been also considered as a strategy to induce a positive effect on the osteoimmune microenvironment [16]. For instance, Mansour et al. [195] reported that by incorporating bone ECM extracts into synthetic dicalcium phosphate (DCP) bioceramics, an increase in the interaction between the biomaterial surface and plasma proteins could be observed in vitro. Furthermore, by coating the scaffolds with ECM molecules enriched with non-collagenous proteins, the levels of the pro-inflammatory cytokines, such as IL-1β, IL-2, TNF-α, and the number of tartrate-resistant acid phosphatase (TRAP) positive osteoclastic cells in a rat tibia model were reduced. In another study, Diez-Escudero et al. [187] compared the effects of biomimetic and sintered CaP scaffolds functionalized with GAGs (ECM molecules capable of binding various growth factors) on the immune response. The covalent functionalization of β-TCP with heparin (a member of the GAG family) led to a reduced pro-inflammatory immune response coupled with the enhanced osteogenic potential of MSCs when incubated with conditioned media harvested from the macrophages grown in contact with the functionalized surfaces. However, by functionalizing a calcium-deficient hydroxyapatite (CDHA) substrate with heparin, an enhancement in the osteoclastic activity and function coupled with a resorptive process of the substrate could be observed.

Finally, the incorporation of nucleic acids, such as siRNA, pDNA or microRNA, into implantable biomaterials emerged as an attractive strategy to induce local specific cell responses [196]. The local delivery of microRNA is especially advantageous due to the fact that these molecules can inhibit various pathways with minimal immunogenicity. For example, Mencia-Castano et al. [197] functionalized a collagen–nanohydroxyapatite composite construct with a microRNA-133a inhibitor and the results showed enhanced bone deposition in a calvaria defect in rats due to the stimulation of the CD206^+^ M2 macrophages’ recruitment.

Altogether, based on the bone metabolism mechanism, various osteoimmunomodulatory cytokines and bioactive molecules have been incorporated into different bone biomaterials with the purpose of modulating the macrophage polarization state and signaling pathways, to regulate the osteogenic process directly and indirectly. Even though the presented results are helpful for the achievement of a desirable osteoimmune environment, further studies are still needed.

### 5.2. Immunomodulatory Biomaterials for Osteoclastogenesis

The successful implantation of an orthopedic biomaterial requires the orchestrated activation and function of the two main types of bone cells, namely osteoblasts and osteoclasts [270]. Therefore, an understanding of the mechanisms through which bone-resorptive osteoclasts interact with the biomaterial could lead to the design of suitable implant surfaces capable of modulating the osteoclastogenesis towards desirable results [271]. However, the activation or deactivation of the osteoclastogenic process as an important regulator of bone regeneration and remodeling has only been recently highlighted [13] in a series of studies [135,141,241,271,272,273,274,275,276,277,278,279,280,281,282,283,284,285,286,287,288,289,290,291,292,293,294,295,296,297,298].

#### 5.2.1. Modification of the Surface Chemistry

Tuning the different surface chemical characteristics of the biomaterial’s surface can represent a strategy employed to modulate the activation and function of osteoclasts. First of all, it was found that highly hydrophilic surfaces are capable of decreasing osteoclast activity and function by inhibiting macrophage adhesion and fusion into FBGCs. Bang et al. [278] studied the effect of two different Ti surfaces (sandblasted/acid-etched and hydrophilic sandblasted/acid-etched Ti) on osteoclastic differentiation and function. Their results showed that the hydrophilic surface was capable of downregulating important osteoclastic markers such as TRAP, c-FOS, osteoclast-associated immunoglobulin-like receptor (OSCAR) and NFATc1. Furthermore, ion immobilization could lead to a suppressed osteoclastogenic process. Bose et al. [281] reported that by doping a β-TCP substrate with Mg^2+^, reduced activity in the resorptive process could be observed. Similarly, studies on RAW 264.7 macrophage cells seeded on a single TCP substrate or Mg^2+^-functionalized TCP substrate demonstrated that the modified surface inhibited osteoclast differentiation and actin ring formation. In addition, many studies have reported that one of the most commonly used materials for bone regeneration, namely calcium phosphate ceramics, can influence the osteoclastogenic process. This class of ceramics includes materials with different chemical compositions and crystal phases, such as α- and β-TCP and HA [282]. However, despite their high potential as bone substitutes, the bioceramic materials present a series of disadvantages that limit their use in the bone regeneration field. For instance, HA is scarcely absorbed within the body, leading to a permanent stress concentration and poor stability [283], while β-TCP possesses a high degradation rate due to its high solubility, poor mechanical properties and insufficient osteoinductivity and osteogenicity [284]. On this basis, in order to overcome the limitations imposed by these materials, biphasic calcium phosphate bioceramics consisting of both β-TCP and HA in varying ratios have been fabricated [271]. Wepner et al. [286] studied the effect of the newly developed electrospun biphasic HA/β-TCP nanoscaffolds (ratio 40/60) on human osteoblasts (hFOB 1.19) and monocytes (THP-1) and the results revealed that both cell lines showed no cytotoxic effect, reduced apoptosis and well-differentiated osteoclast-like cells. Similarly, Yamada et al. [285] evaluated the effect of various calcium phosphate ceramics (HA, β-TCP and HA/β-TCP substrates with different composition ratios) on neonatal rabbit bone cells in terms of resorptive activity. The obtained results showed that the HA/β-TCP substrate with a ratio of 25/75 elicited the most extensive resorption activity, while the single β-TCP substrate presented only small discontinuous resorptive lacunae and no resorptive lacunae on the HA or HA/β-TCP (ratio 75/25) substrates. Moreover, the incorporation of silicon (Si) into HA substrates led to higher osteoclastic activity compared to the single HA substrate [287].

Altogether, the osteoclastogenic process can be influenced by surface chemical modifications, with a downregulated osteoclastic differentiation on hydrophilic surfaces.

#### 5.2.2. Modification of the Physical Properties

Another strategy used to modulate various cell functions is represented by surface topography modification. Surface roughness is an important modification method used in order to modulate processes such as osteogenesis and osteoclastogenesis and to ensure rapid bone integration with the implantable biomaterial. In terms of Ti implants, various methods, such as polishing and sandblasting, have been used in a number of studies [291,298]. Brinkmann et al. [291] studied the effects of various Ti surfaces (smooth (TS), acid-etched (TA) and sandblasted/acid etched (SLA)) on RAW 264.7 macrophages. Their findings demonstrated that osteoclasts on rough surfaces (TA and SLA) showed a similar osteoclastogenic process to those observed on natural bone tissue, whereas on the smooth surfaces, the osteoclastogenesis was limited. Similarly, Sommer et al. [292] investigated the osteoclastogenesis process on various substrates (Ti, TiAl6Mo7, CoCr28Mo6 and FeCrNi) with sandblasted and polished surfaces and the results revealed that the number of osteoclasts and TRAP activity were higher on the rough surfaces in comparison to the smooth surfaces. However, the osteoclastogenic process was not significantly affected by the various alloying compositions. Ion et al. [289] investigated the behavior of the RAW 264.7 cell line on mesoporous nanochannels generated in hot glycerol–phosphate electrolyte on the surface of a Ti50Zr alloy. The acquired results showed that the nanochannels led to a decrease in the proliferation rate and levels of pro-inflammatory cytokines released into the culture media. In addition, the nanochannels supported cell adhesion but did not permit macrophages to undergo fusion and form FBGCs. Furthermore, in a further study [290], the same nanochannelar surface induced an inhibition of the differentiation and maturation of osteoclasts from their precursors when treated with RANKL. Regarding calcium phosphate ceramics, Davison et al. [241] modified the macrostructure of a biphasic calcium phosphate substrate and evaluated its effect on the RAW 264.7 cell line. The results showed that the substrate with a smaller surface microstructure led to the formation of ectopic bone and multinucleated osteoclast cells in dog muscles. However, the substrates with a larger surface architecture formed neither new bone tissue nor osteoclast-like cells. Costa et. al. [280] deposited on PCL surfaces HA coatings with various submicrometer and micro-scale topographical characteristics, with the purpose of investigating the effect of various roughness levels of HA on the bone-forming cells’ behavior. The results demonstrated that the micro-rough HA coating inhibited the resorption ability of the osteoclasts isolated from the long bones of New Zealand white rabbits, in comparison to the smoother HA coating, which presented resorption lacunae. This inhibitory effect could be explained by the presence of disruptions in the F-actin sealing zones observed only on the micro-rough surfaces. Collectively, these studies suggest that nano/microscale surface architecture could represent a major feature in osteoclastogenesis modulation. However, future studies are needed in order to fully elucidate the exact mechanism through which surface architecture affects the osteoclast activity and function.

#### 5.2.3. Loading of Various Cytokines and Biological Molecules

Recently, numerous methods for surface functionalization that modulate the osteoclasts’ activity, especially as an osteoporosis treatment, have been employed. The immobilization of alendronate, an anti-osteoporosis drug, represents one of the most commonly studied approaches. Lee et al. [294] evaluated the in vitro and in vivo effects of gold nanoparticles (GNPs) functionalized with alendronate on bone marrow macrophages (BMM). The results acquired following in vitro investigation revealed that the osteoclast activity was significantly reduced in comparison to the control group. Moreover, the in vivo results showed enhanced bone regeneration, similar to that of the host bone. Similarly, Boanini et al. [279] investigated the influence of alendronate-coated mesoporous glass nanospheres on the osteoclast activity and their findings demonstrated reduced osteoclastogenic activity. In another study, Forte et al. [295] immobilized both alendronate and quercetin on an HA substrate and the results revealed a reduction in the osteoclast viability and differentiation capacity on the functionalized HA substrate, as compared to the control group. Furthermore, Boanini et al. [279] studied the influence of a quercetin-functionalized HA substrate on human osteoclast precursor cells, T-110, and the in vitro results suggested that the conjugated quercetin was capable of reducing the level of secreted cytokines and the osteoclast activity.

Electrospun fibers are promising delivery platforms due to the technique’s ability to produce fibers with properties similar to those of the ECM, thereby allowing them to receive various biological molecules or drugs. Ghag et al. [296] functionalized PCL electrospun fibers with poly(vinyl phosphonic acid-co-acrylic acid) (PVPA-AA) in order to investigate the effects of the newly developed PCL/PVP A-AA scaffold on human osteoblast cells (HOBs) and human osteoclast precursor cells. The results obtained demonstrated a combined positive effect of the newly developed scaffold on both osteoblast and osteoclast behavior. Therefore, favorable osteoblastic cell differentiation and maturation coupled with an enhanced mineralization process was reported. On the other hand, the presence of the PVPA-AA polymer led to a reduction in the number of osteoclasts due to an upregulation in the expression of OPG. Moreover, Riccitiello et al. [297] prepared electrospun nanofibers from resveratrol (RSV), a drug used to influence the osteoclastogenic process, and the results showed that the newly synthesized fibers suppressed the maturation of the osteoclast precursors. In another study, Negrescu et al. [235] coated an AZ31 Mg alloy with electrospun PCL fibers loaded with coumarin (CM) and/or ZnO NP, and the in vitro results showed that the co-presence of the ZnO NP and CM in the coatings reduced the differentiation and formation of mature osteoclasts.

Despite the aforementioned studies, it can be stated that the number of biologically active molecules used for surface functionalization in terms of modulating the osteoclastogenesis process is quite limited. Therefore, more research has to be conducted in this respect in order to develop biomaterials via functionalization with biomolecules exhibiting high potential to treat bone defects.

## 6. Conclusions and Future Perspectives

The success of an orthopedic implant is determined by how well the biomaterial integrates into the in vivo local bone microenvironment and is capable of modulating the bone-healing cascade events. However, the response of the host immune system triggered by biomaterial implantation is one of the most significant critical issues that needs to be overcome for the development of bone implants. It is a well-known fact that the early inflammatory state represents the first step in the natural healing process, and that the chronic inflammatory state that appears as a response to biomaterial implantation can lead to impaired bone regeneration and, ultimately, implant failure. It comes not as a surprise that novel biomaterials that exhibit excellent biocompatibility and are capable of sustaining cell viability are also highly immunogenic, being capable of eliciting an inflammatory reaction. The traditional implant-designing methods are focused on developing biocompatible biomaterials capable of suppressing the FBR and, implicitly, the excessive inflammation and fibrous tissue encapsulation of the implant. However, in recent years, a more in-depth understanding of the bone biology and the influence of the immune system on the function of the bone-forming cells led to a new direction in research and the emergence of a new field called “osteoimmunology”. It was found that immune cells participate actively in bone dynamics, both under physiological and pathological conditions, through the release of various regulatory molecules such as cytokines, transcription factors, growth factors and signaling molecules. Therefore, given the importance of the immune system cells in bone dynamics, a shift in the paradigm of the nature of biomaterials has been considered. In consequence, the design methods do not focus on fabricating only “inert” (e.g., biocompatible) biomaterials but also “immunomodulatory” implants, able to induce and modulate a favorable immune response rather than suppressing it. The type of immune response elicited by the host is dictated by the properties of the biomaterial, which can be modified in order to obtain different features capable of influencing the protein adsorption and signaling factors’ binding. Numerous studies have reported different surface modification strategies, such as chemical and physical alteration and functionalization with biological molecules. The chemical tuning of the biomaterial surface with various functional groups, surface charges or modifications to improve the physical properties, such as topography and stiffness, can be effectively employed to regulate the functions of the immune and bone-forming cells. Moreover, the incorporation of various biological molecules can also be used as an approach to modulate the immune functions and bone metabolism. Likewise, a significant number of studies have suggested the important role of macrophage phenotype modulation, an approach which has been widely employed even though the exact mechanism is still unclear, and it requires further investigation. It is well known that both the inflammatory reaction and the wound healing process are intimately connected to changes in the redox balance, and even though, at low concentrations, oxidative stress exhibits various physiological roles, an upregulation of ROS production and persistence over a long period of time can prove to be harmful to the host. Moreover, low concentrations of ROS can positively modulate the macrophage polarization state towards an M2 phenotype, while a negative effect on the phenotype switch has been observed in different studies when ROS production has been inhibited. In order to stimulate new bone formation and modulate the immune response induced by ROS production, various natural antioxidants, such as quercetin, resveratrol, curcumin, etc., have been included in different biomaterials/scaffolds with positive results. In addition, compared to macrophages, information about the role and potential of other immune cells, such as dendritic cells or T cells, in the bone remodeling process is limited in the current literature. Another pressing matter is represented by the limited number of studies regarding the influence of the biomaterials on osteoclast activity and function. The existing literature suggests that the osteoclastogenic process is modulated by regulatory molecules secreted by the immune cells and osteoblasts. However, the majority of studies have reflected the in vitro situation, with only a few studies approaching the in vivo biomaterial-mediated osteoclastogenesis.

Despite the enormous progress that has been made in the field of osteoimmunomodulatory biomaterials for bone regeneration, the need for further studies that can clarify the exact interplay between immune and bone-forming cells and help to develop functional biomaterials capable of inducing a proper material–host response is compelling.

## Figures and Tables

**Figure 1 materials-14-01357-f001:**
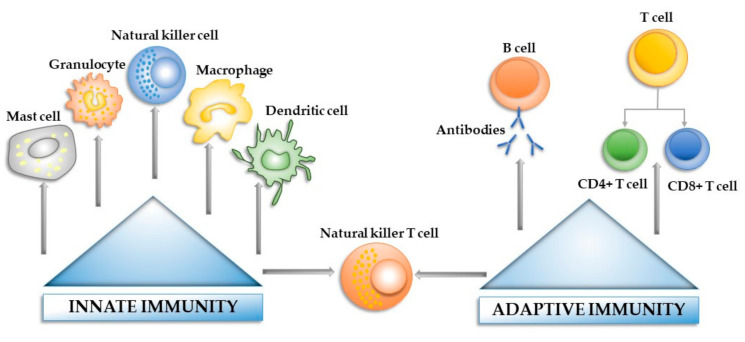
The main components of the innate and adaptive immune systems.

**Figure 2 materials-14-01357-f002:**
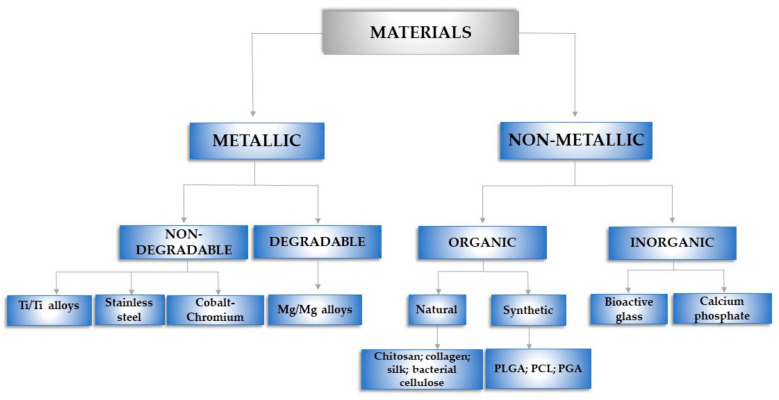
Various materials used in the bone tissue engineering field. PGLA: poly(lactic-co-glycolic acid); PCL: poly-ε-caprolactone; PGA: poly(glycolic acid).

**Figure 3 materials-14-01357-f003:**
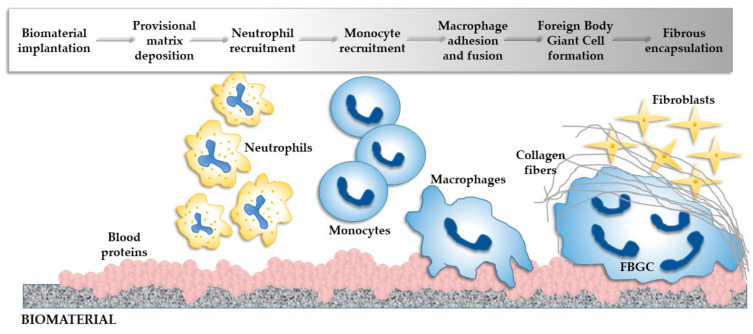
The sequence of events contributing to the immune response towards biomaterial implantation.

**Figure 4 materials-14-01357-f004:**
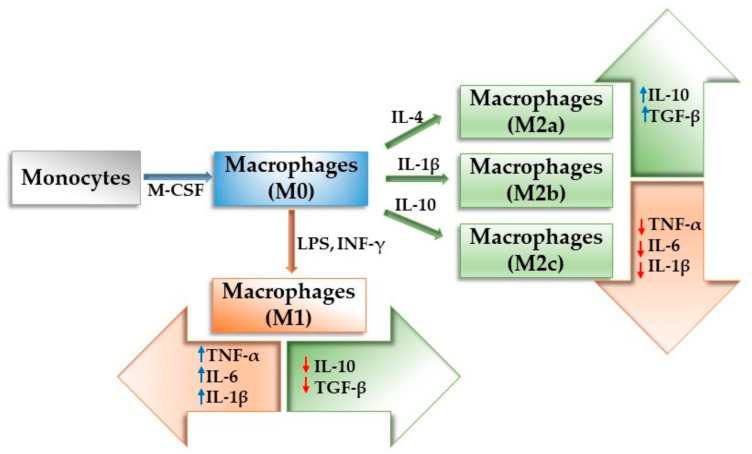
Polarization of macrophages and the released cytokines. ↑ indicates upregulated production; ↓ indicates downregulated production; IL—interleukin; TGF-β—transforming growth factor β; TNF-α—tumor necrosis factor α; INF-γ—interferon γ; LPS—lipopolysaccharide.

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
