# Peer review of "The State of the Art and Prospects for Osteoimmunomodulatory Biomaterials"

_materials, 2021, doi:10.3390/ma14061357_

Round 1

Reviewer 1 Report

I think it would not be superfluous to add to the review information about the term "osteoimmunology" which was first used in 2000 (Kiselevsky MV, Anisimova N.Yu., Martynenko NS, Sitdikova SM, Dobatkin SV, Karaulov AV, Estrin Yu.Z. OSTEO IMMUNOLOGY AND BIOCJMPATIBILITY OF BONE IMPLANTS.Immunology. 2018; 39 (5-6). DOI: http://dx.doi.org/10.18821/0206-4952-2018-39-5-6-305-311)
Line 172. I recommend that the authors add bacterial cellulose to the list of materials (Films of Bacterial Cellulose Prepared from Solutions in N-Methylmorpholine-N-Oxide: Structure and Properties. Processes 2020, 8, 171. https://doi.org/10.3390/pr8020171 )
Line 173. "Various biomaterials used in the bone tissue engineering field" The caption must be corrected. Steel is not a biomaterial ... 

Author Response

We would like to thank the Reviewer 1 for taking the time to review our manuscript and for the constructive comments which led to an improvement of the work. We have paid close attention to each point raised by the Reviewer and corrected the manuscript by considering his/her comments. Please find hereinafter the responses to the raised questions.              

1. I think it would not be superfluous to add to the review information about the term "osteoimmunology" which was first used in 2000 (Kiselevsky, M.V.; Anisimova, N.Y.; Martynenko, N.S.; Sitdikova, S.M.; Dobatkin. S.V.; Karaulov, A.V.; Estrin, Y.Z. OSTEOIMMUNOLOGY AND BIOCOMPATIBILITY OF BONE IMPLANTS. Immunology 2018, 39 (5-6), http://dx.doi.org/10.18821/0206-4952-2018-39-5-6-305-311).

Response: We kindly took into consideration the Reviewer’s suggestion and introduced in the revised form of the manuscript additional information regarding the term “osteoimmunology”. However, we were not able to find nor access the recommended article with any of the scientifically data research engines, therefore the information added has been synthesized from on other topic articles. Please see the paragraph between lines 71-76.

2. Line 172. I recommend that the authors add bacterial cellulose to the list of materials (Films of Bacterial Cellulose Prepared from Solutions in N-Methylmorpholine-N-Oxide: Structure and Properties. Processes 2020, 8, 171, https://doi.org/10.3390/pr8020171).

Response: We kindly took into consideration the Reviewer’s indication and we added a modified figure that includes the bacterial cellulose (please see Fig. 2, page 5) as a material used in the bone regeneration field and added the suggested paper to the References list (ref. 52) the suggested paper.

3. Line 173. "Various biomaterials used in the bone tissue engineering field" The caption must be corrected. Steel is not a biomaterial.

Response: Thank you for this notice, we have rectified the figure caption according to the suggestion. Please see line 186.

Reviewer 2 Report

Broad comments

The authors belong to a research group with expertise in the field. The manuscript is well written with organized content and only some minor irregularities. It is exhaustive on the topic, it will be useful for PhD students and interdisciplinary teams starting working with bone biomaterial development or application of them in regenerative medicine. However, such paper will likely deter many from reading because it is currently too long and some parts are more suitable to a book chapter rather than a focused review. With the suggested revision the manuscript can be improved, thus highlighting the demand of the osteoimmunology field and pushing the demand of novel biomaterials regulating osteoclast activity, thus some chronic disease.

Major comments

Abstract is more than 300 words. Author instruction tells max 200 words.

ALTERNATIVE ABSTRACT

The critical role of the immune system in host defence against foreign bodies and pathogens has been long recognized. With the introduction of a new field of research called osteoimmunology, the crosstalk between the immune and bone forming cells has been studied more thoroughly, leading to the conclusion that the two systems are intimately connected through various cytokines, signalling molecules, transcription factors and receptors. The host immune reaction triggered by biomaterial implantation determines the fate of the implant. The traditional biomaterial design consisted in fabricating inert biomaterials capable of stimulating the osteogenic process. however, in vitro and in vivo results were reported. By endowing the orthopaedic biomaterials with favourable osteo- immunomodulatory properties, a desired immune response can be triggered in order to obtain a proper bone regeneration process. In this context, various approaches such as the modification of chemical/structural characteristics or the incorporation of bioactive molecules, have been employed in order to modulate the crosstalk with the immune cells. The current review provides an overview of recent developments in such applied strategies.

  • Section 2 introduces the immune system to researchers outside life science and medicine, but should be recapped. For example
  1. Lines 114-119 can be removed.
  2. Line 138-153, may be removed. Indeed, the adaptive response as represented by B–cells and T-cells did not actively participate in the second part of the review.
  3. Please try to condense some parts.

  • Section 3 has to be shortened. I suggest as following.
  1. Summarize the coagulation cascade events, leaving only what is important for to understand the following sections and not what is universally known at basic level.
  2. The wound healing process has relevance, however I suggest to restrict to the most important events and players.

  1. Current sections, 4.1 “Macrophage origins” and 4.2. “Macrophage polarization states” can be joined. The 373-389 are considered as histology and not concerning osteodifferentiation. Lines 390-393, I believe, are not relevant to this article. I highly suggest to remove the section and title “Macrophage origins” , while saving just lines from 365-371 and 393-399. Give your new appropriate title to the joined new section.

  1. Currently, there are 2 sections named 4.2, but amending the comment above you will have just one. Please check in the end the order of section titles.
  2. Remove 478-479. Not useful.
  3. In most of the cases, in table 1 the column “Effect of immune cells” can be optimized, for example diving in 2 subcolumns (e.g. physicalproperty1; effect1 _ physicalproperty2;effect2). Please, also be more schematic and not textually descriptive
  4. Except for the immune cell effects, the hydrophicility reduces in general cell attachment to the implant. If the authors agree with this, I suggest mentioning it very briefly. Or please reply to explain to me and just discuss it, thank you.
  5. From line 799 to line 901 the topic is the biomaterial modifications in titanium or titanium coating. Please try to group the findings. The authors can remove some of the oldest referenced works or reducing details.
  6. Knowing the role of ROS in inflammation state, immune cell function and biomaterial relationship, the role of oxidative stress before and after the implant is critical and mentioned by the authors. In this context, the authors may mention (perhaps in the concluding section 6) the novel strategies in order to counteract excessive inflammation as well as modulate the oxidative stress in situ, such as the incorporation of antioxidant molecules into the biomaterial (please see doi.org/10.3390/bioengineering7030104). Indeed, many natural compounds (e.g. the quercetin or resveratrol mentioned in the 5.2.3 by the authors) are able to change chemo-physical properties and have immunomodulatory effects.
  7. How did the authors, in the huge literature about the topic, select the articles that are nicely reporting the biomaterials in section 5? May you create a small flow chart or description and put this info in a supplementary file. Do the most convenient thing for you, but let the reader aware of it.
  8. In 5.1.1. , The authors should add a small table summarizing the Ions (Mg, Si, Zn, etc) and the effects with references. By doing it, the text in the section can be reduced as well as the final manuscript wourd count.
  9. “Surface chemistry alterations” – “Physical properties alternations” – “Delivery of cytokines and biological molecules” are repeated twice (like subparagraphs, I guess the author meant that). However, it is better to make a small edit and distinct titles in 5.1 from titles in 5.2. Also useful, for MDPI full-text html summary
  10. First sentence of current 5.2.3 section can be easily deleted without loosing info. (line 1131).

Minor comments

  1. A good review has been very recently published, “Biocompatibility of nanomaterials and their immunological properties” online at doi: 10.1088/1748-605X/abe5fa. I believe it could be useful for the readers and in line with the current manuscript to make a relevant loop of quality studies.
  2. Please define extending the name of “MSCs” at line 472. They have been never abbreviated so far.
  3. Authors wrote “Osteogenesis is one of the two bone forming major processes Osteogenesis is one of the two bone forming major processes”, but what did they mean for the second one? Bone regeneration or osteoclastogenesis? I guess it’s right to specify that is bone regeneration. Please be careful in suggesting osteoclastogenesis as bone forming process. Literally it means differentiation towards osteoclasts (as you already know by writing lines 543-545) and only bone loss (resorption) can be directly associated with. Please answer this point if I misinterpreted.
  4. Line 543, typo , The *osteoclasts are vital players(instead of the singular)
  5. Line 702, typo, biomaterial’
  6. Line 709 BMSCs is used with the same meaning of different similar abbreviation above, please make uniform. On line 846 there is BMCs
  7. Line 788 typo, alternations
  8. Line 1052, typo? biomaterial’s surface
  9. Line 1060. MFTATc1, please define the abbreviations. Take care of all biological abbreviations please, to be understood by non-biologists also.
  10. Line 1118, the comma ?

Author Response

The authors would like to thank the Reviewer 2 for taking the time to review this manuscript and for the constructive comments which led to an improvement of our work. We have paid close attention to each point raised by the Reviewer and corrected the manuscript by considering his/her comments. Please find hereinafter the responses to the raised questions.

Major comments

1. Abstract is more than 300 words. Author instruction tells max 200 words.

Response: Thank you so much for this notice and for the alternative abstract which we highly appreciate. Please see the Abstract section (lines 9-28).

  • Section 2 introduces the immune system to researchers outside life science and medicine, but should be recapped. For example:

1. Lines 114-119 can be removed.

Response: Thank you very much for the thorough analysis of our paper. We kindly took into consideration the Reviewer’s recommendation and removed the information between the lines 114-119. Please see lines 120-125.

2. Line 138-153, may be removed. Indeed, the adaptive response as represented by B–cells and T-cells did not actively participate in the second part of the review.

Response: We thank you for the useful recommendation. The information between the lines 138-153 has been summarized in order to retain only the most vital information. Please see lines 145-167.

3. Please try to condense some parts.

Response: We kindly took into the account the Reviewer’s indication and the information from Section 2 has been better summarized. Please see pages 3-4.

  • Section 3 has to be shortened. I suggest as following.

4. Summarize the coagulation cascade events, leaving only what is important for to understand the following sections and not what is universally known at basic level.

Response: Thank you for the recommendation, we kindly took it into consideration and we summarized the coagulation cascade, with only keeping the essential information. Please see lines 216-257.

5. The wound healing process has relevance, however I suggest to restrict to the most important events and players.

Response: We kindly took into consideration the Reviewer’s indication and reformulate the information concerning the wound healing process in a way that it retains only the important events, without losing its coherence.  Please see lines 335-375.

6. Current sections, 4.1 “Macrophage origins” and 4.2. “Macrophage polarization states” can be joined. The 373-389 are considered as histology and not concerning osteodifferentiation. Lines 390-393, I believe, are not relevant to this article. I highly suggest to remove the section and title “Macrophage origins”, while saving just lines from 365-371 and 393-399. Give your new appropriate title to the joined new section.

Response: Thank you for pointing this out and for the constructive comments. We have removed Section 4.1. regarding the macrophage origins and replaced it with a reformulated section (4.1. Macrophage plasticity and polarization) that includes the remaining lines and the previous section 4.2. Please see lines 392-452.

7. Currently, there are 2 sections named 4.2, but amending the comment above you will have just one. Please check in the end the order of section titles.

Response: Thank you for this notice and the constructive comments.

8. Remove 478-479. Not useful.

Response: We kindly took into consideration the Reviewer’s indication and we eliminated the information between lines 478-479. Please see lines 515-517.

9. In most of the cases, in table 1 the column “Effect of immune cells” can be optimized, for example diving in 2 subcolumns (e.g. physicalproperty1; effect1 _ physicalproperty2;effect2). Please, also be more schematic and not textually descriptive.

Response: Thank you for the recommendation. We have modified the table according to the suggestion and the newly obtained table can be found in the revised manuscript (page 15).

10. Except for the immune cell effects, the hydrophilicity reduces in general cell attachment to the implant. If the authors agree with this, I suggest mentioning it very briefly. Or please reply to explain to me and just discuss it, thank you.

Response: We agree with the Reviewer and, in line with this, it is shown in Table 1 (page 15) that “hydrophilicity reduces macrophage adhesion” and mentioned between the lines 713-716 the following sentence “… an increased hydrophilicity results in a high protein adsorption resistance which can lead to decreased interactions with the immune cells, which in turn may reduce the immunomodulatory effects” [refs. 53,82, 151-160]. To note that, it is a general trend that mammalian cells interact better with the moderately hydrophilic surfaces, not highly hydrophilic where there is the risk of the water interposing between the material surface and the adsorbed protein layer. A possible explanation for these results could be represented by the preferential adsorption of cell-adhesive proteins onto the moderately hydrophilic surfaces.

Considering that the surface wettability is the major factor in promoting cell attachment/ guiding the subsequent cell behavior, several studies focused on its influence on cell adhesion. However, conflicting results have been reported mostly due to the use of materials characterized by different surface topographic parameters such as roughness, micro-texture..., and different surface chemistry that affect the material wetting behavior. Thus, a large body of literature reports greater cell attachment and cell spreading on hydrophilic positively/negatively charged modified surfaces in comparison to hydrophobic surfaces [Langmuir 2004, 20, 11684–11691; Biomaterials 2006, 27, 3096–3108; J. Biomater. Appl. 2011, 26, 327-347; Mater. Sci. Eng. C 48, 2015, 365-371, etc.]. On the other hand, some research groups have reported that the cells adhered and proliferated at the highest rate when cultured on various hydrophobic surfaces, but not superhydrophobic ones [Biomaterials 2003, 24, 4621–4629; Hao, L. et al. Laser Surface Treatment of Bio-implant Materials, 2006, Great Britain, Wiley; J. Biomed. Mater. Res. 2009, 90A, 133–141; Science China 2017, 60, 614-620].

Moreover, the contradictory results could be explained by the different cellular types investigated that may exhibit different cell surface receptors, particularly integrin types interacting with specific extracellular molecules. Being non-adherent or semi-adherent, it is expected that the immune cells show a different behavior when interact with the underlying substrate, as compared to the adhesion-dependent cells..

11. From line 799 to line 901 the topic is the biomaterial modifications in titanium or titanium coating. Please try to group the findings. The authors can remove some of the oldest referenced works or reducing details.

Response: We kindly took into the consideration the Reviewer’s recommendation and we removed the unnecessary details regarding the studies presented in between the mentioned lines. This information followed a logical scheme in which studies between lines 847-885 contained information about the influence of the surface roughness/microroughness on the tissue regeneration process; studies between lines 886-927 summarized information about nanoscaled biomaterials; studies between lines 928-953 contained information about nanoscale coatings and patterns.

12. Knowing the role of ROS in inflammation state, immune cell function and biomaterial relationship, the role of oxidative stress before and after the implant is critical and mentioned by the authors. In this context, the authors may mention (perhaps in the concluding section 6) the novel strategies in order to counteract excessive inflammation as well as modulate the oxidative stress in situ, such as the incorporation of antioxidant molecules into the biomaterial (please see doi.org/10.3390/bioengineering7030104). Indeed, many natural compounds (e.g. the quercetin or resveratrol mentioned in the 5.2.3 by the authors) are able to change chemo-physical properties and have immunomodulatory effects.

Response: Thank you for the recommendation and for the suggested article. We kindly took into consideration the suggestion and added in the revised manuscript a detailed description regarding the novel strategies employed to counteract severe inflammatory responses by incorporation of natural antioxidants into different biomaterials/scaffolds, which can be found in the concluding Section between lines 1261-1271. Furthermore, the suggested article has been added to the Reference list (ref. [73]).

13. How did the authors, in the huge literature about the topic, select the articles that are nicely reporting the biomaterials in section 5? May you create a small flow chart or description and put this info in a supplementary file. Do the most convenient thing for you, but let the reader aware of it.

Response: The authors thank the Reviewer for this suggestion. The logic that stood behind the article selection for the present review has been explained through a flowchart which can be found in the Supplementary Materials (Fig. 1S).

14. In 5.1.1. , The authors should add a small table summarizing the Ions (Mg, Si, Zn, etc) and the effects with references. By doing it, the text in the section can be reduced as well as the final manuscript word count.

Response: We kindly took into consideration the Reviewer’s recommendation and we have added a new table in which we have summarized the effect of the ions released from the implantable devices on the immune response and bone related events. Please see Table 2 (page 18).

15. “Surface chemistry alterations” – “Physical properties alternations” – “Delivery of cytokines and biological molecules” are repeated twice (like subparagraphs, I guess the author meant that). However, it is better to make a small edit and distinct titles in 5.1 from titles in 5.2. Also useful, for MDPI full-text html summary.

Response: Thank you for pointing this out. We have modified the titles of the subsection 5.2.1, 5.2.2 and 5.2.3. The new titles can be found at lines 1103, 1144 and 1184, respectively.

16. First sentence of current 5.2.3 section can be easily deleted without losing info. (line 1131).

Response: We kindly took into the consideration this recommendation and we have removed the suggested line (Please see lines 1185 and 1186).

Minor comments

17. A good review has been very recently published, “Biocompatibility of nanomaterials and their immunological properties” online at doi: 10.1088/1748-605X/abe5fa. I believe it could be useful for the readers and in line with the current manuscript to make a relevant loop of quality studies.

Response: We thank you for this recommendation but we found ourselves in the incapacity to access the article and the information contained in it except for the Abstract section.

18. Please define extending the name of “MSCs” at line 472. They have been never abbreviated so far.

Response: Thank you for this notice, this abbreviation is spelled at its first appearance into the main text (line 509).

19. Authors wrote “Osteogenesis is one of the two bone forming major processes Osteogenesis is one of the two bone forming major processes”, but what did they mean for the second one? Bone regeneration or osteoclastogenesis? I guess it’s right to specify that is bone regeneration. Please be careful in suggesting osteoclastogenesis as bone forming process. Literally it means differentiation towards osteoclasts (as you already know by writing lines 543-545) and only bone loss (resorption) can be directly associated with. Please answer this point if I misinterpreted.

Response: We thank you for this notice. Indeed, we were referring to bone regeneration, and not osteoclastogenesis and the suggested modification in the manuscript has been made, which can be found at line 519.

20. Line 543, typo, The *osteoclasts are vital players (instead of the singular).

Response: Thank you for pointing out this mistake. We corrected this typo (Please see line 583).

21. Line 702, typo, biomaterial’.

Response: Thank you for this notice, the correct word is spelled at lines 278, 746 and 1104 (answer to Q24).

22. Line 709 BMSCs is used with the same meaning of different similar abbreviation above, please make uniform. On line 846 there is BMCs.

Response: We kindly thank you for pointing this. We made sure to use the same abbreviations (BMSCs) throughout the whole manuscript. Please see lines 880 and 897.

23. Line 788 typo, alternations.

Response: Thank you for pointing this, we have modified the mistake accordingly. Please see line 836.

24. Line 1052, typo? biomaterial’ssurface.

Response: We kindly thank you for this notice, we have rectified the mistake. Please see line 1104.

25. Line 1060. MFTATc1, please define the abbreviations. Take care of all biological abbreviations please, to be understood by non-biologists also.

Response: Thank you for pointing this. We have noticed that there was a mistake in the abbreviated name of the molecule (the correct one is NFATc1), that has been rectified (line 1112) and the associated explanation (nuclear factor of activated T-cells, cytoplasmatic 1) has already been presented in the original manuscript. Please see line 611.

26. Line 1118, the comma?

Response: We warmly thank you for this notice. The correction has been made (line 1171).

Round 2

Reviewer 2 Report

The authors demonstrated a very kind and professional conduct in welcoming my advices and performing a thorough revision. They addressed all the raised points excellently.

Sorry if we have put you in a hurry work, but…compliments!